

# Divergent evolutionary histories of DNA markers in a Hawaiian population of the coral *Montipora capitata*

Hollie M. Putnam[1,2,*], Diane K. Adams[3,*], Ehud Zelzion[4], Nicole E. Wagner[4], Huan Qiu[4], Tali Mass[5], Paul G. Falkowski[3,6], Ruth D. Gates[1] and Debashish Bhattacharya[4]

[1] Hawai'i Institute of Marine Biology, Kaneohe, HI, United States of America
[2] Department of Biological Sciences, University of Rhode Island, Kingston, RI, United States of America
[3] Department of Marine and Coastal Sciences, Rutgers University, New Brunswick, NJ, United States of America
[4] Department of Ecology, Evolution, and Natural Resources, Rutgers University, New Brunswick, NJ, United States of America
[5] Marine Biology Department, University of Haifa, Haifa, Israel
[6] Department of Earth and Planetary Sciences, Rutgers University, Piscataway, NJ, United States of America
[*] These authors contributed equally to this work.

## ABSTRACT

We investigated intra- and inter-colony sequence variation in a population of the dominant Hawaiian coral *Montipora capitata* by analyzing marker gene and genomic data. Ribosomal ITS1 regions showed evidence of a reticulate history among the colonies, suggesting incomplete rDNA repeat homogenization. Analysis of the mitochondrial genome identified a major (*M. capitata*) and a minor (*M. flabellata*) haplotype in single polyp-derived sperm bundle DNA with some colonies containing 2–3 different mtDNA haplotypes. In contrast, *Pax-C* and newly identified single-copy nuclear genes showed either no sequence differences or minor variations in SNP frequencies segregating among the colonies. Our data suggest past mitochondrial introgression in *M. capitata*, whereas nuclear single-copy loci show limited variation, highlighting the divergent evolutionary histories of these coral DNA markers.

Corresponding author
Debashish Bhattacharya,
debash.bhattacharya@gmail.com

## INTRODUCTION

Coral reef ecosystems are centers of marine biodiversity that provide a number of ecological services, including food, income from tourism, nutrient cycling and waste removal, and absorption of wave energy to mitigate erosion (*Sheppard et al., 2005*). Integral to the success of the coral holobiont is the complex and intimate interplay between the animal cnidarian host and one or more types of symbiotic dinoflagellate algae, as well as their microbiomes comprised of prokaryotes and viruses (*Meyer & Weis, 2012*; *Bhattacharya et al., 2016*; *Röthig et al., 2016a*). These complex biotic interactions are thought to confer a variety of properties, including the ability to tolerate stress and adapt to changing environments (*Röthig et al., 2016b*). Disturbance of these associations can lead to the death of the coral host (*Haas et al., 2014*).

The adaptive ability of corals is also encoded in their genomes. In addition to the DNA polymorphisms expected in outbreeding diploids, corals have multiple additional sources of genetic variation. These include intra- and inter-species chimerism (*Puill-Stephan et al., 2009*; *Work et al., 2011*; *Schweinsberg et al., 2014*; *Rinkevich et al., 2016*); in *Acropora millepora*, both molecular marker analysis and direct observation demonstrate juvenile fusion during settlement with gregarious larvae forming chimeric colonies (*Puill-Stephan et al., 2009*). Hybridization between species has been demonstrated between corals in the field and in the lab, but has a disputed and potentially rare contribution to genetic diversity. There is also extensive genetic evidence for historic introgression in corals, putatively through hybridization, but post-zygotic barriers may limit its contribution to genomic evolution except in marginal habitats (reviewed in *Willis et al., 2006*). Another source of variation is the maintenance of multiple copies of genes either through heteroplasmy of organelles or replication of genes within the genome. Furthermore, genetic variation can accumulate through mutations in somatic tissues (mosaicism) that differentiate in individual polyps (*Van Oppen et al., 2011*; *Schweinsberg, Tollrian & Lampert, 2016*), propagate in the somatic tissue, and may be transferred to subsequent generations if the germline is not segregated (*Work et al., 2011*; *Schweinsberg et al., 2014*). However, an independently segregating germline has been suggested in the coral *Orbicella faveolata*, which could protect gametes from propagating these mutations (*Barfield, Aglyamova & Matz, 2016*). Finally, genome-wide analyses have demonstrated that horizontal gene transfer (HGT) occurs in corals with ca. 0.2% of the animal gene inventory comprised of foreign genes (*Bhattacharya et al., 2016*). The primary functions of these genes are to expand existing stress response pathways such as those involved in DNA repair and protection against reactive species (*Bhattacharya et al., 2016*). Previous work has also shown that corals and sea anemones acquired a pathway via HGT that produces photo-protective mycosporine amino acids that absorb UVR (*Shinzato et al., 2011*). These data suggest that coral colonies may be characterized as dynamic hubs of genetic variation that allow them to respond to changing environmental conditions (e.g., *Schweinsberg, Tollrian & Lampert, 2016*). This hypothesis remains however to be tested using complete genome data from a local population.

Here we looked in detail at the nature and sources of genetic variation in the coral host within a restricted set of colonies. Specifically, our goal was to determine the contribution of current and past chimerism and hybridization using coral sperm samples (from egg/sperm bundles) from Kāneohe Bay, O'ahu, Hawai'i. To this end, we compared individual sperm bundles from a single polyp and pooled sperm from multiple polyps using traditional marker genes, non-coding regions, and novel genomic data. Our results demonstrate widely contrasting outcomes when using standard markers such as ribosomal ITS1 and the mitochondrial DNA control region (MTC) in comparison to single-copy genes identified in a genome-wide analysis. The MTC data provide evidence for past mitochondrial introgression, whereas the nuclear data indicate genetic uniformity.

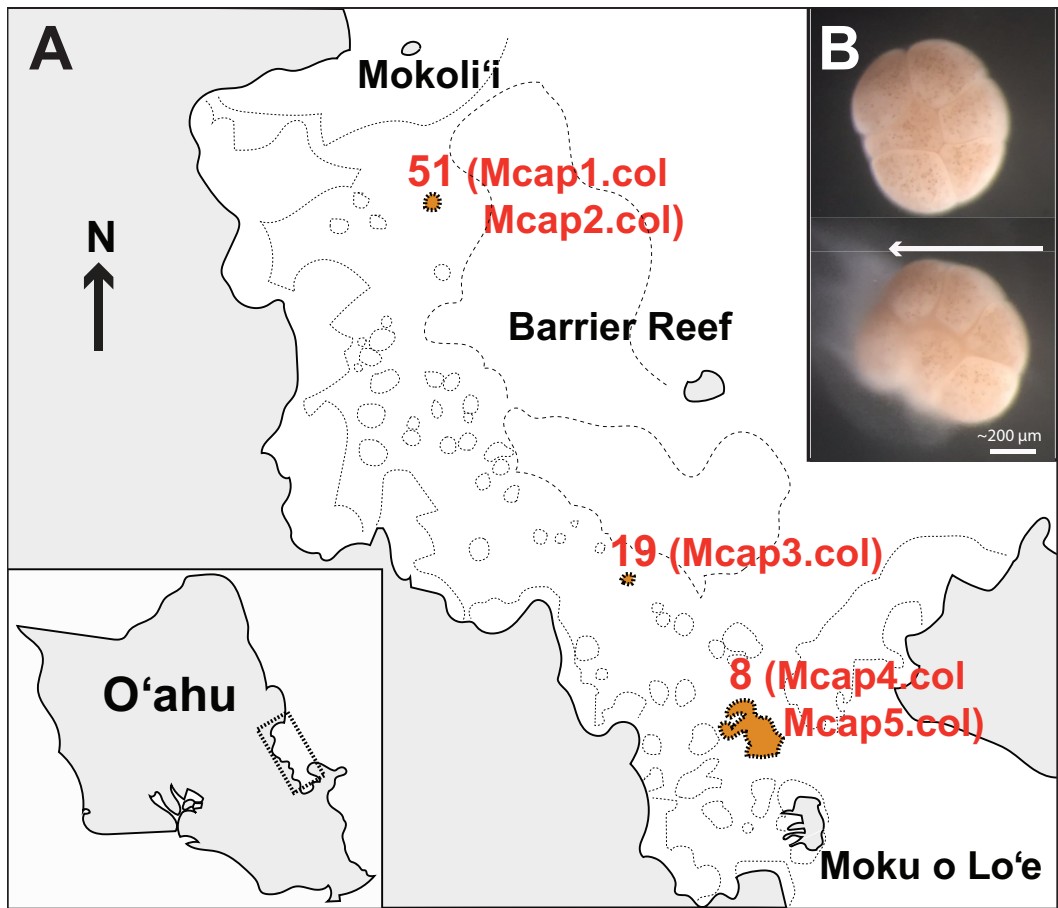

**Figure 1** **Sampling plan for *Montipora capitata* in Hawai'i.** (A) The location of the three reefs and five colonies sampled for this study in Kāneohe Bay, O'ahu. (B) The release of a single bundle of *M. capitata* eggs (large cream sphere) and sperm (cloudy pool below eggs) from an individual polyp (scale bar shown). This sperm tissue was isolated from the eggs and used to generate the single sperm bundle and pooled (from multiple polyps) marker gene, genomic, and transcriptomic data.

## MATERIALS AND METHODS

### Coral collection and holding

*M. capitata* corals were collected on June 2015 (Special Activity Permit 2015–17) from Kāneohe Bay, O'ahu Hawai'i (Fig. 1A) and returned to the Hawai'i Institute of Marine Biology where they were housed in outdoor 1,300 L tanks in shaded conditions with an ambient photoperiod and natural diurnal fluctuations in temperature (Fig. S1). The sources of tissues used in our study are as follows (see Fig. 1A): Reef 51 (colonies Mcap1 and Mcap2), Reef 19 (colony Mcap3), and Reef 8 (colonies Mcap4 and Mcap5).

In the July 2015 spawning period, we collected 10 individual egg sperm bundle replicates from different regions of each of the five colonies as well as a mixture of bundles from each colony (Table S1 shows the sample sequencing plan for each molecular marker). Individual egg sperm bundles were collected to allow us to discriminate between variation within a polyp (e.g., somatic mutations, genetic duplications) and within a colony (e.g., chimerism).

Egg sperm bundles were collected immediately upon release and placed in 1.5 mL sterile RNAse and DNase free microfuge tubes. The bundles were left in individual tubes for 30 min to break apart from each other, with the buoyant eggs floating to the surface and the denser sperm (Fig. 1B) settling to the bottom. The sperm fraction was removed by pipetting to a new tube and was cleaned by a series of three rinse-and-spin steps, with samples rinsed with 0.2 µm filtered seawater and centrifuged at 13,000 rpm for 3 min. The supernatant was removed and the concentrated sperm was stored at −80 °C. To generate a gene inventory for the downstream transcriptomic analysis, sperm was collected during the same spawning period (June 2015) from five different field netted adults of *M. capitata* on the fringing reefs on the west side of Moku o Lo'e (Fig. 1A) and used for RNA extraction.

## DNA extraction and genomic shotgun library construction

Genomic DNA was extracted from individual sperm bundles using the Zymo Quick-DNA Universal Kit (Zymo Research, Irvine, CA, USA), with the Biological Fluids and Cells protocol, and eluted in 50 µL of 10 mM Tris–HCl (pH 8.5). DNA concentrations were measured on a Qubit instrument. A total of 200 ng of genomic DNA from the single sperm bundle Reef 51 Mcap2.bundle9 was used to construct a library using the Illumina TruSeq Nano DNA LT Library Prep Kit (Illumina, Inc., San Diego, CA, USA). We chose colony Mcap2 that had limited variation in the ITS1 and MTC trees (see Results below) to reduce polymorphisms that could complicate genome assembly. The library was run on an Illumina MiSeq Personal Genome Sequencer using the Illumina MiSeq Reagent Kit v3 (600 cycles, paired-end).

## RNA extraction and RNA-seq library construction

Total RNA from sperm bundles collected from the five field colonies was extracted by resuspending each sample in 550 µL of Trizol (ThermoFisher Scientific) (*Mass et al., 2016*). These samples were passed twice through a QiaShredder column (Qiagen, Inc., Hilden, Germany) and then transferred to a 1.5 mL microcentrifuge tube. A total of 450 µL of Trizol was added to bring the volume to 1.0 mL. Following a 5-minute room temperature incubation, 200 µL of chloroform was added and the sample was vigorously shaken for 15 s, and then incubated at room temperature for 3 min. The samples were centrifuged for 15 min at 4 °C and the upper aqueous layer was transferred to a new 1.5 mL tube, and an equal volume of 70% ethanol was added and gently mixed. The samples were transferred to Qiagen RNeasy mini columns. From here onwards, the Qiagen RNeasy mini protocol was followed, including the optional on-column DNase treatment. Total RNA was eluted in 55 µL of nuclease-free water. Five individual RNA-seq libraries were generated using 200 ng of the total RNA from each sample using the Illumina TruSeq RNA Library Preparation Kit v2. The libraries were combined in equimolar concentrations and run on a single Illumina MiSeq flowcell using the Illumina MiSeq Reagent Kit v3 (150 cycles, paired-end). The genomic and transcriptomic data created for this project are available under NCBI BioProject PRJNA339779.

## Cloning and Sanger sequencing

The MTC region was amplified from sperm DNA using the primers Ms_FP2 (5′-TAG ACA GGG CCA AGG AGA AG-3′) and MON_RP2 (5′GAT AGG GGC TTT TCA TTT GTT TG-3′) (*Schweinsberg, Tollrian & Lampert, 2016*). The ITS1 region was amplified using the primers ZITS1 (5′-TAA AAG TCG TAA CAA GGT TTC CGT A-3′) and ZITS2 (5′-CCT CCG CTT ATT GAT ATG CTT AAA T-3″) (*Forsman et al., 2009*). The PCR was done with Platinum HiFi Taq (ThermoFisher Scientific, Waltham, MA, USA) for the multiple bundle samples (denature 2 min, 94 °C, followed by 35 cycles of 30 s at 94 °C, 30 s at 55 °C, and 30 s at 68 °C, and finished with a 5 min 68 °C extension), and with the NEBNext High-Fidelity 2X Master Mix (ThermoFisher Scientific, Waltham, MA, USA) for the single bundle samples (denature 3 min, 95 °C, followed by 35 cycles of 10 s at 95 °C, 30 s at 55 °C, and 30 s at 72 °C, and finished with a 5 min 72 °C extension). Amplicons were purified using Beckman Coulter AMPure XP beads, and Sanger-sequenced using the same primers as for PCR amplification.

Some of these MTC and ITS1 amplification products were cloned into the vector pCR-Blunt II-TOPO using the Invitrogen Zero Blunt TOPO PCR Cloning Kit (ThermoFisher Scientific). Ten colonies from each set were picked, plasmids were purified, and inserts were Sanger-sequenced using the vector-specific primers SP6 and T7. We also amplified a widely used *Pax-C* intron using the primers Mont_Pax-FP1 and Mont_Pax-RP1 (*Van Oppen, Koolmees & Veron, 2004*). For this region, we used a 3 min, 98 °C denature step, followed by 35 cycles of 98 °C 30 s, 55 °C 30 s, 72 °C 30 s, and finished with a 5 min 72 °C extension, using the NEBNext High-Fidelity 2X Master Mix.

Using sequences of putative single-copy genes identified in the *M. capitata* RNA-seq assembly PCR primers were designed to yield ca. 200–500 bp genomic DNA amplicons (Table 1; see below), assuming no intron sequences in the regions of interest. These single-copy genes were amplified using PCR and multiple bundle sperm DNA from colonies Mcap1, Mcap2, and Mcap3 with the NEBNext High-Fidelity 2X Master Mix, with a 3 min, 98 °C denature step, followed by 35 cycles of 98 °C 30 s, 65 °C 30 s, 72 °C 30 s, and finished with a 5 min 72 °C extension. PCR reactions were cleaned using the Qiagen QiaQuick PCR Purification Mini Kit; then, one-half of each cleaned amplicon sample was run on a 0.8% agarose TAE gel, the DNA bands were excised and extracted using the Qiagen QiaQuick Gel Extraction Kit and eluted into 30 μL of 10 mM Tris-HCl (pH 8.5). The amplicons were Sanger-sequenced using the same primers as for amplification.

## Transcriptomic and genomic data generation and analysis

The RNA-seq run with the combined five sperm libraries yielded 36,250,700 MiSeq raw reads that were adapter and quality-trimmed using the CLC Genomics Workbench (v7.5, Qiagen, Inc., Hilden, Germany). After trimming, 22,681,438 reads (3.2 Gbp of data) remained for assembly and downstream analysis. The trimmed reads were assembled with CLC Genomics Workbench into 73,094 contigs with a N50 = 442 bp. The *M. capitata* genomic DNA library was run twice on the MiSeq and yielded a total of 95,971,984 raw reads, of which 70,060,127 were used for assembly, producing 600,706 contigs totaling 359,691,707 bp and with a N50 = 720 bp.

Putnam et al. (2017), *PeerJ*, DOI 10.7717/peerj.3319

Peer*J*

**Table 1** Transcriptome-derived single-copy gene markers used for the SNP/indel analysis of *M. capitata* colonies.

| *M. capitata* transcriptome contig | Query length (amino acids) | Annotation in *Acropora digitifera* | Target length (DNA) | Number of SNPs/ indels | SNP/indel frequency | PCR primers used for genomic DNA amplification |
|---|---|---|---|---|---|---|
| 760 | 126 | grpE protein homolog 1, mitochondrial-like | 227 | 0 | – | F (5′-ACAAGGAAGCAGATAGAGGAAGCC-3′), R (5′-CAGCCACAGTTCCTGCCTCC-3′) |
| 2,660 | 223 | protein angel homolog 2-like | 554 | 3 | 53.7 | F (5′-AAGCGGATGGCTGTGCTACT-3′), R (5′-AGCATCCCCGTGCAACAAAGT-3′) |
| 3,032 | 159 | ubiquitin-conjugating enzyme E2Q-like protein 1 | 248 | 0 | – | F (5′-CCTGACAACTTCCCATTTGCCC-3′), R (5′-ACCCTCAGATTTTGGCGGTGT-3′) |
| 4,432 | 204 | uncharacterized protein LOC107336263 | 242 | 0 | – | F (5′-GAAGCCGCCGAGCGTAGAAT-3′), R (5′-AATTTGCGTCGCGGAGGTGA-3′) |
| 16,812 | 211 | protein ABHD17B-like | 287 | 0 | – | F (5′-CAAAGTTGGCGTTCCTGCCC-3′), R (5′-AAGAATGACGCCCGCACACT-3′) |
| 26,437 | 158 | upstream stimulatory factor 1-like isoform X2 | 378 | 0 | – | F (5′-CCGACCCAGGAACGCTTCAA-3′), R (5′-TCTGTGCCTCGGCCATTGTG-3′) |
| 26,632 | 181 | E3 ubiquitin-protein ligase NRDP1-like | 322 | 1 | 40.5 | F (5′-TGTGGAGGTGTGCTGGAGGA-3′), R (5′-GGTCGTCAATCTGGGCCTGC-3′) |
| 32,815 | 214 | methylmalonic aciduria and homocystinuria type C protein homolog | 251 | 2 | 45 | F (5′-TCAGGTGGTGCACAAGGCAT-3′), R (5′-GGGTGCACACTCACTCCACA-3′) |
| 62,797 | 200 | dopamine receptor 2-like | 385 | 0 | – | F (5′-TGCTGTGGGGCGTTCTTAGC-3′), R (5′-CGAGCTGTTCATCACCGGCA-3′) |

## Single-copy gene analysis

To identify single copy genes for the analysis of single nucleotide polymorphisms (SNPs), we constructed orthologous gene families using proteomes from seven anthozoan species (Table S2) with OrthoFinder under the default settings (BLASTP search $e$-value $\leq 1e-5$; MCL inflation $I = 1.5$) (*Emms & Kelly, 2015*). A total of 1,632 gene families were found to be present in single copy in all studied species. We used genes from the coral *Stylophora pistillata* as queries to look for single-copy gene homologs in the *M. capitata* genome and transcriptome assemblies using TBLASTN ($e$-value cut-off $< 10^{-10}$). In the transcriptome data, we found 489 hits to single-copy genes, but only 9 of them (see Table 1) had $>20\times$ average coverage when mapping the *M. capitata* DNA reads to the transcriptome assembly (parameters: 90% identity over 50% of the read length). The SNP/indel calling was done using the CLC Genomics Workbench. To insure the high quality of the SNPs/indels, the SNP/indel phred score was set to $\geq 20$ (i.e., 1% error rate) and the flanking 3 nucleotides set to $\geq 15$ (i.e., ca. 5% error rate). In the genome assembly, we found 463 single-copy genes, but because of its highly fragmented nature, the vast majority of the hits were scattered among more than one genomic contig. Only eight partial single-copy genes (Table S3) were found to reside on one contig per gene. The RNA-seq reads from the combined 5-colony data set (see above) were mapped to these eight genomic contigs (parameters: 80% identity over 80% of the read length), which guided the manual gene model construction. Using these gene models, we again mapped the genome data to them to identify SNPs/indels in both coding and non-coding regions with the same parameters used for the transcriptome analysis. Therefore, in total we studied SNPs in 17 single-copy genes in *M. capitata* by mapping genomic reads to assembled transcripts (nine genes) and to assembled genomic data for an additional 8 genes.

## RESULTS

### PCR analysis of ITS1, MTC, and *Pax-C*

We tested two specific hypotheses about limited, local populations with the molecular marker data produced in this study: (1) no differences exist in the host genotype of multiple colonies ($n = 3$ colonies for ITS1 and MTC, and $n = 5$ colonies for *Pax-C*), and (2) no differences exist in the host genotype within a colony ($n = 10$ polyps in three colonies for ITS1 and MTC; see Table S1 for the sample naming scheme). Our approach was to isolate sperm DNA from pooled sperm bundles from multiple *M. capitata* polyps from colonies Mcap1, Mcap2 (Reef 51), and Mcap3 (Reef 19), and as egg/sperm bundles of individual polyps from Mcap2 (e.g., see Fig. 1B), respectively. These DNAs were used as templates for PCR-amplification of the ribosomal ITS1 region. Sanger sequence analysis of the PCR products derived from the pooled sperm bundles showed overlapping peaks on the chromatograms, therefore we cloned individual PCR products. Ten ITS1 clones from each of the three-targeted coral colonies were sequenced (e.g., Mcap1.col.c1-c10 (col = the contribution from a mixture of polyps in each colony)) and the manually aligned 546 nt was used to generate a maximum likelihood tree (PhyML; GTR + I model of evolution (alignment available as File S1)). We also added ITS1 data from individual sperm bundle

DNAs isolated from different polyps in colony Mcap2 (numbered Mcap2.bundle1-10.pcr (bundle = the contribution from a single polyp in each colony)) (Table S1). Inclusion of NCBI top-hit ITS1 data from different *Montipora* species (as outgroup) in this tree shows that the *M. capitata* sequences form a well-supported monophyletic lineage (100% bootstrap) that includes existing data from this species (shown in violet text). The ITS1 regions from the different colonies are however intermingled and form 4–5 different clades (albeit many with weak bootstrap support). This topology suggests that individual polyps in the same colony contain distinct ribosomal operons (i.e., with ITS acting as the marker). For example, in Mcap2 (brown text in Fig. 2A), there is a dominant ITS1 form; i.e., all 10 single polyp sperm DNAs contain this sequence, whereas from the cloned multiple sperm bundles, 6/10 are also identical with the single polyp data, with the remaining four encoding distinct sequences. Mcap1 contains ITS1 sequences (shown in sienna text) that are located in at least 5 different clades, and similar results are found for Mcap3 (shown in blue).

To gain another perspective on the ITS1 data and account for the poor bootstrap support of many nodes in the ITS1 tree, we used SplitsTree4 (*Huson & Bryant, 2006*) to generate a network with the Net approach using all sites and the GTR + I model of evolution. Here, we excluded the non-*M. capitata* data, the Mcap2 single bundle data, and combined all identical or nearly identical (single SNP-bearing) sequences to simplify the analysis. This network (Fig. 2B) represents uncertainty in branches that connect nodes as parallel lines and shows that the evolutionary relationships are highly unresolved, consistent with a complex evolutionary history for these rDNA gene families in the different sperm DNA samples. The Phi test in SplitsTree provides significant ($p = 0.004$) support for recombination among these ITS1 regions.

Analysis of the maternally encoded MTC marker (466 nt) (*Work et al., 2011*) included multiple cloned sequences from colonies Mcap1-3 as well as a MTC sequence identified in Mcap2.bundle9 shotgun genomic library sequencing (see below). PhyML analysis of these data (Fig. 2C) shows that mtDNA has a complex evolutionary history in these corals with several *Montipora* species (e.g., *M. danae*, *M. verrucosa*) forming a monophyletic clade and the cloned MTCs representing at least 3 *M. capitata* haplotypes (alignment available as File S2). NeighborNet analysis provides no evidence for reticulate evolution among these sequences (Fig. S2, phi test $p = 0.8068$). The *M. flabellata* MTC region is distinct from the *M. capitata* clade. These results suggest that multiple mtDNA haplotypes occur in the three studied *M. capitata* colonies.

Given these results with ITS1 and MTC, we targeted the spliceosomal intron that is located between exons 46 and 47 in the nuclear encoded single-copy homeobox gene *Pax-C* (*Van Oppen et al., 2000*). Analysis of *Pax-C* intron PCR products from pooled bundles from all five colonies (i.e., including Mcap4 and Mcap5, Table S1) also showed a single SNP in this region. The PhyML tree includes the top 100 BLASTN hits to the *Pax-C* single-copy intron region we isolated from the Kāneohe Bay population (Fig. 3) (alignment available as File S3). Despite the fact that this region encodes a single SNP in the studied colonies, its placement in the tree shows significant polyphyly of not only *M. capitata*, but also many other species in this genus.

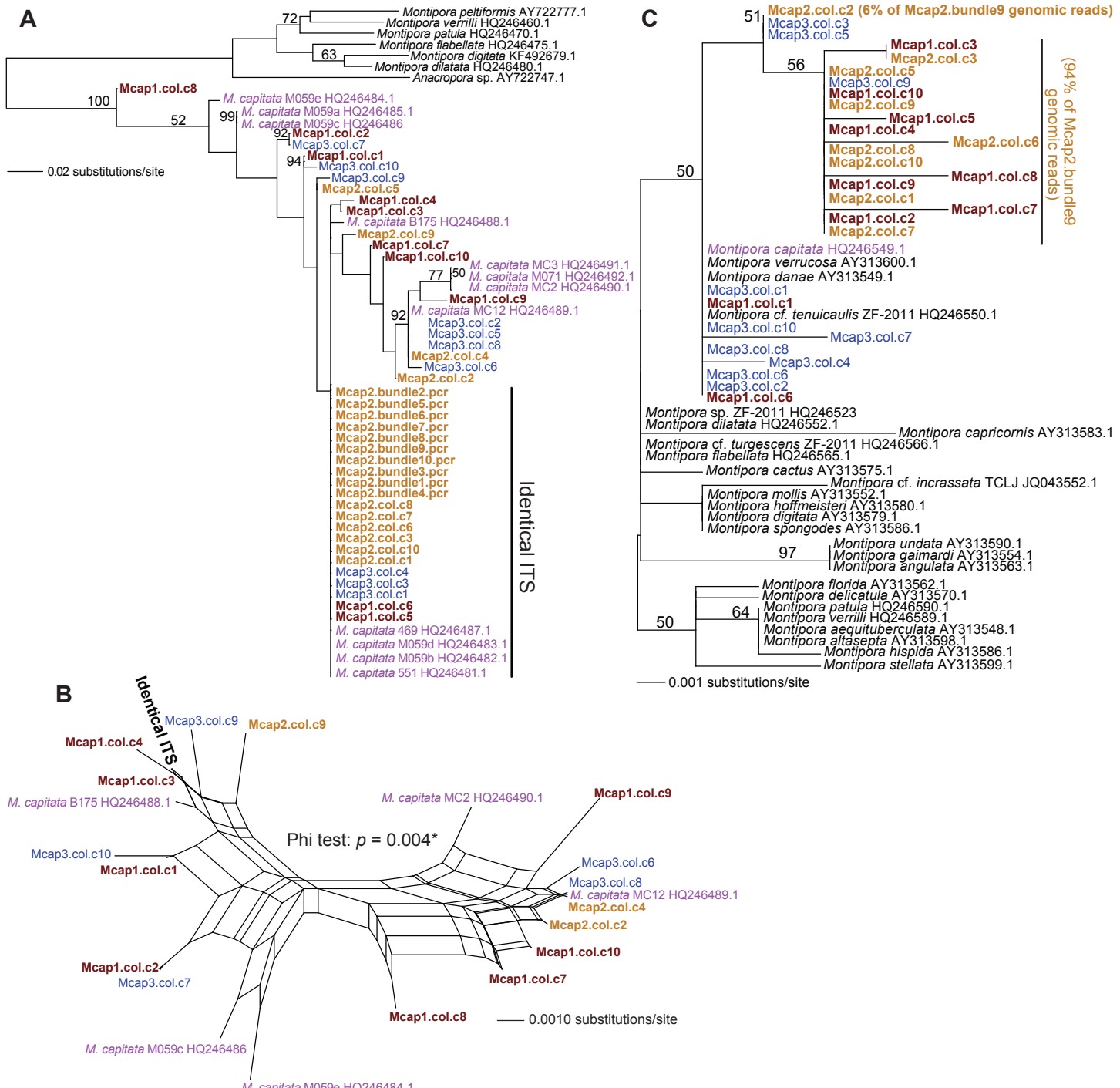

**Figure 2** **Phylogenetic analysis of ITS1 and MTC data from *M. capitata*.** (A) PhyML tree of ITS1 showing the distribution of cloned sequences from three colonies that were harvested for multiple bundle sperm DNA (Mcap1, Mcap2, and Mcap3 in sienna, brown, and blue text, respectively). Existing *M. capitata* data are shown in violet text and other species in black. Bootstrap values (100 replicates) are shown at the nodes. The cloned single sperm bundle sequences from Mcap2.col.c1-10 that are identical with each other and with other ITS1 data are marked. (B) Results of the NeighborNet analysis of the ITS1 data. (C) PhyML tree of the MTC data. The colony number demarcation and support values are the same as in (A). The clades containing the dominant (94%) and minor mitochondrial haplotypes (6%) in the Mcap2.bundle9 genomic data are marked.

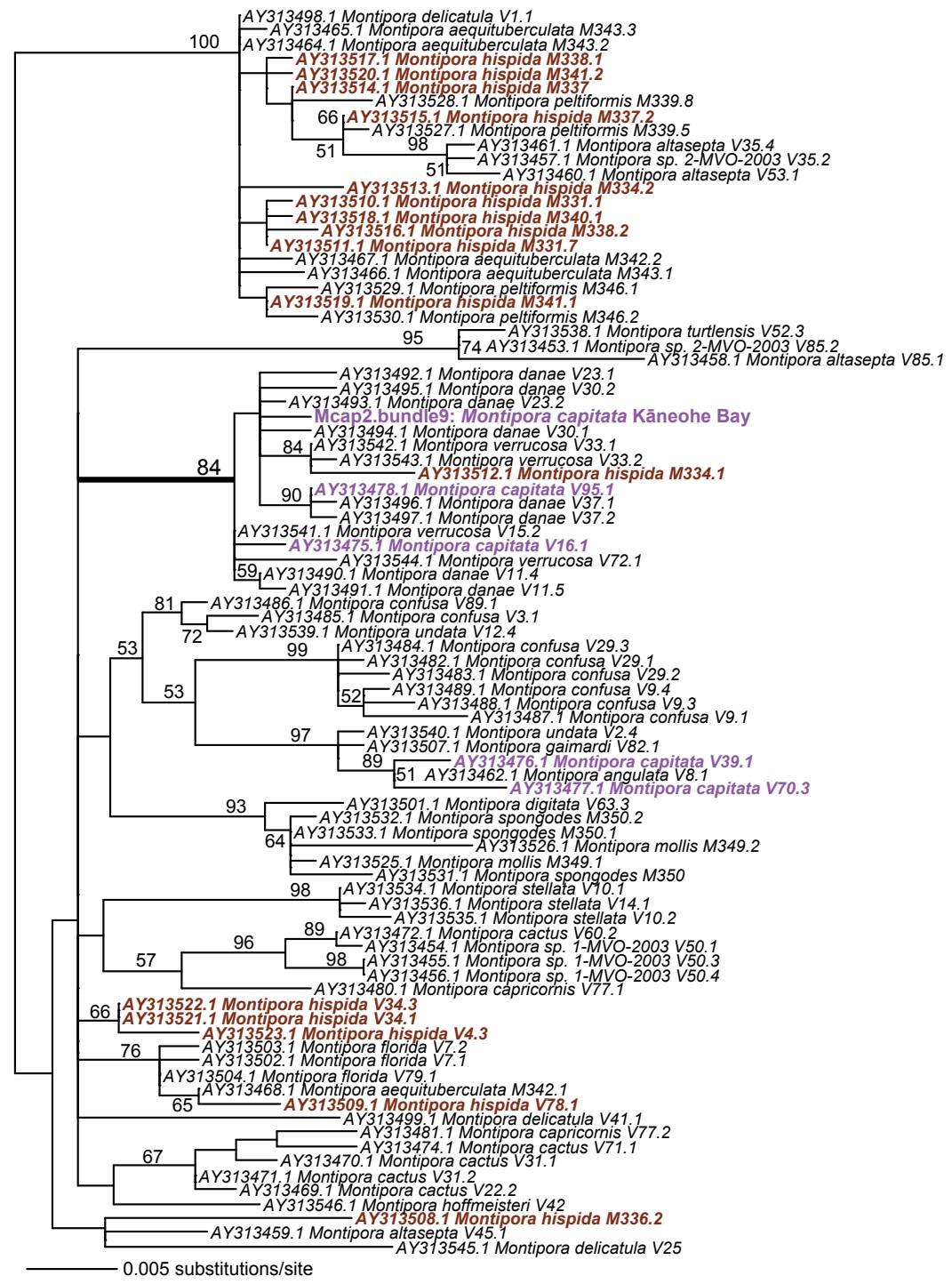

**Figure 3** PhyML tree of the partial *Pax-C* intron region that includes the sequence from Mcap2.bundle9 and the top 100 BLASTN hits from GenBank. The *Montipora capitata* isolates are shown in purple text and *M. hispida* isolates in sienna. Bootstrap values (100 replicates) are shown at the nodes.

## Genomic analysis of ITS, MTC, and *Pax-C*

We analyzed draft genome data (14 Gbp of Illumina sequence with an average read length = 200 bp) from the single sperm bundle DNA isolated from Mcap2.bundle9 (Table S1). The ITS1 PCR fragment from this DNA was Sanger sequenced and provided clean data although some sites showed minor peaks supporting low frequency A–G substitutions. This ITS1 sequence was used to recruit genomic reads (80% identity over 80% of the read length) from the Mcap2.bundle9 data for alignment and SNP calling. This analysis showed that 19 high quality SNPs/indels were present among the 6,394 reads that mapped to the partial ITS1 region (615 bp). Within these mapped reads, the SNP/indel frequencies were low, varying from 2.2% to 15.6% at polymorphic sites. These results suggest that the rDNA ITS1 region in Mcap2.bundle9 is represented by a nearly homogenous collection of sequences, with some minor variation among repeat copies.

For the MTC genomic analysis, we again used a Sanger sequence of the MTC region from Mcap2.bundle9 region (716 bp) to recruit genomic reads (726 total). This analysis turned up 9 high quality SNPs/indels varying from a frequency of 4.9%–18% in the MTC region. Of these SNPs/indels, 6% of the reads in one region encoded an 8 nt insertion that co-segregated (i.e., in the same reads) with a G-rich region lacking 5 nt, one upstream SNP, and one downstream single nucleotide insertion (Fig. 2C and Fig. 4). A second region of mtDNA near the MTC that shows this type of variation is shown in Fig. S3. These data suggest that, in addition to the minor variation observed in this mtDNA region, at least two haplotypes can be resolved in the Mcap2.bundle9 single sperm bundle data with one occurring in low frequency; i.e., ca. 6%. When this MTC sequence was used to query NCBI, the top five hits shared 99% identity (5 SNPs) with the query, of which two are from different isolates of *M. flabellata* and one each from *M.* cf. *turgescens*, *M. dilatata*, and *M. turtlensis*. When the higher frequency (94%) MTC region was used as the query, the top five hits were identical and derived from two different isolates of *M. capitata*, whereas the other three are different isolates of *Montipora* sp. ZF-2011. These results are consistent with the presence of two different mtDNA haplotypes in Mcap2.bundle9 DNA. A more complete analysis of mtDNA variation was achieved by collecting all genomic reads that mapped (90% identity over 50% of the read length) to the existing *Montipora cactus* mtDNA (NC_006902.1). These reads were used to generate a consensus *M. capitata* mtDNA assembly that represents a mixture of at least two genomes (therefore not shown here) with the dominant sequence (present at >50% at individual positions) used to identify each site in mtDNA. This reference was used to map the genome data (as above) and identified, including the regions shown in Figs. 4 and S2, 109 polymorphic regions with regard to SNPs and indels in the *M. capitata* data (see Table S4).

We identified the *Pax-C* intron in the draft assembly and mapped 31 Mcap2.bundle9 genomic reads to the 5′-terminus of the region (419 bp). The results of this analysis are shown in Fig. S4A and demonstrate the presence of one SNP at the frequency of 50% confirming the result from the Sanger sequenced PCR products. The *Pax-C* intron PCR product from Mcap2.bundle9 produced a clean sequence with a single unambiguous G to T polymorphism (Fig. S5).

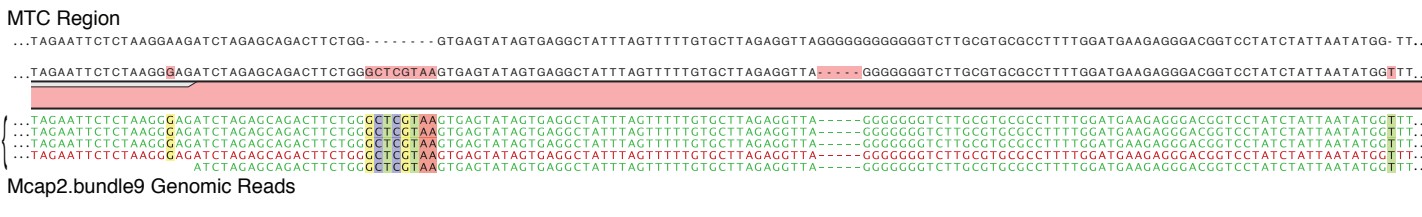

MTC Region

Mcap2.bundle9 Genomic Reads

**Figure 4** **Mapping of Mcap2.bundle9 genomic data to the MTC region (716 bp).** Nine high quality SNPs/indels were found, varying in frequency from 4.9% to 18% in the MTC region. Of these SNPs/indels, 6% of the reads in one region encoded an 8 nt insertion that co-segregated (i.e., in the same reads) with a 5 nt deletion from a G-rich region, a downstream single nucleotide insertion, and one upstream SNP.

## Single-copy gene analysis

Given the conflicting results among the ITS, MTC, and *Pax-C* data, we searched for additional single-copy genes in the *M. capitata* transcriptome data following a stringent procedure (see Methods) and recovered nine candidate partial gene sequences in this assembly with a SNP/indel frequency that ranges from 40.5 to 53.7% (Table 1). We then asked the question whether the low number of SNPS/indels and their frequency would hold if we used PCR to amplify single-copy cDNA regions from multiple bundle sperm cDNA isolated from colony Mcap2, that showed limited variation in the ITS1 and MTC trees (Fig. 2). The SNP numbers and their locations in the genomic mapping of Mcap2.bundle9 sequences were compared to positions of uncertainty (i.e., two coincident strong peaks) in the chromatograms derived from PCR products that encode the same genes (e.g., Fig. S5). These results show that the genomic-based SNP data from Mcap2.bundle9 match exactly the multiple bundle sperm PCR sequence output from this colony.

To study genetic variation in non-coding regions as well as the coding regions, we searched for single-copy genes in the *M. capitata* genome assembly, which were than combined with the relevant RNA-seq data to manually build an additional eight gene models (Table S3). We then mapped the genomic reads back to each genome-derived gene model (e.g., Fig. S4B for the microtubule-associated protein 1A/1B light chain 3C-like sequence (1 SNP) and Fig. S4C for the Myb-like protein X (0 SNPs)) to count the number of SNPs/indels in the coding and non-coding regions. This analysis showed that the eight gene models had either 0 or 1 SNPs, with the frequency of 1 SNP close to 50% (Table S3). The mapping data suggest that the single sperm bundle DNA encodes SNPs at a frequency of ca. 50%, as would be expected for meiotic products derived from a diploid parent. These data suggest that all polyps sampled from Mcap2 are derived from a single genotype with no evidence of mosaicism or chimerism. Analysis of *Pax-C* intron PCR products from all five colonies (i.e., Mcap1-5) showed no novel SNPs in this region (i.e., see Figs. S4 and S5).

## DISCUSSION

Given the existing hypotheses of current and historic coral chimerism (*Rinkevich et al., 2016*; *Van Oppen et al., 2000*; *Puill-Stephan et al., 2012*, e.g., Fig. 5A), mosaicism, and hybridization based on population data (*Schweinsberg, Tollrian & Lampert, 2016*), the major goal of our study was to elucidate the nature and sources of genetic variation in the coral host. Our approach examined the intra- and inter-colony genetic structure in

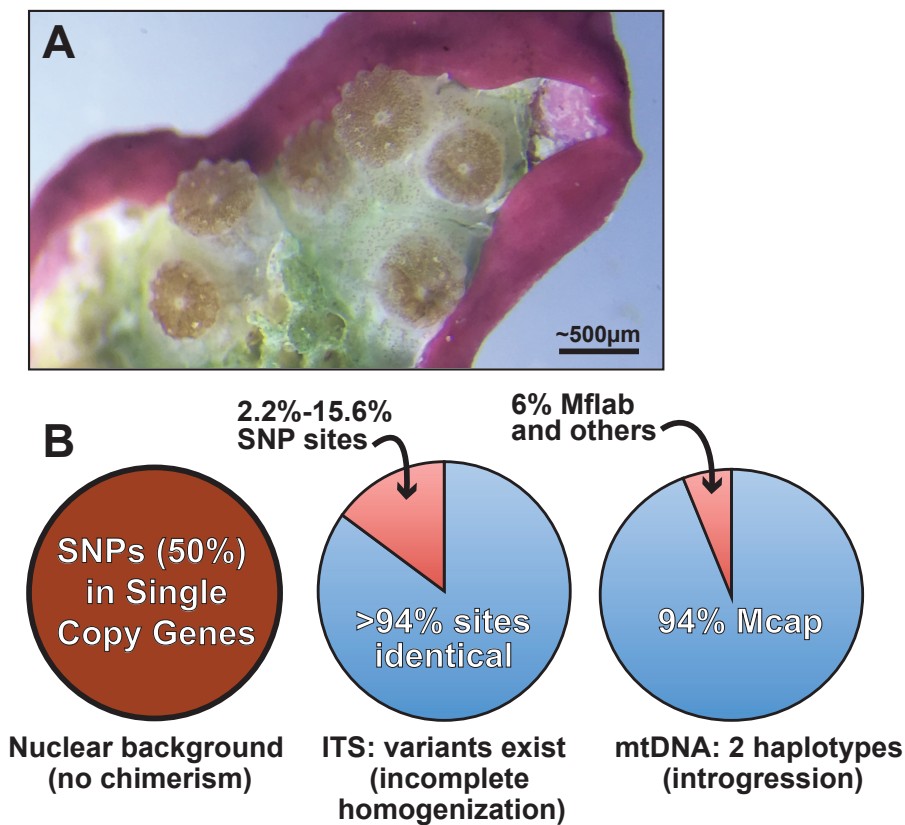

**Figure 5** **Summary of genomic variation in the Kāneohe Bay population of *M. capitata*.** (A) Image of *Montipora* spat that have fused or are near fusion 56 days post-settlement (scale bar shown). These spat all settled individually, but are now in physical contact, as evident in the symbionts below the polyps. Image prepared by Elizabeth Lenz. This type of behavior is thought to underlie *Montipora* MWS and has been observed in many other corals. (B) The results of our analysis of genomic variation in *M. capitata* reefs provide no evidence of chimerism in the nuclear lineage in multiple individual colonies. Significant ITS1 sequence variation is present however, within and between colonies and in genome data from Mcap2.bundle9. We also find two distinct mtDNA haplotypes in the Mcap2.bundle9 genome. Mcap is *M. capitata* and Mflab is *M. flabellata*.

*M. capitata*, which is the dominant reef builder in Hawai'i. This work was made possible by the spawning reproductive strategy of this species that allowed us to isolate sperm (from egg/sperm bundles) from individual polyps. Analysis of a rapidly evolving multi-copy molecular marker, ITS1, revealed a reticulate history among the colonies. Given the inherent issues associated with the use of such complex gene families, the genomic and transcriptomic analyses were used to identify 17 single copy nuclear loci. Analysis of sequence variation in these loci, from combined genomic and RNA-seq data, or Sanger sequencing data, displayed minimal genetic variation in the sampled *M. capitata* (summarized in Fig. 5). These findings of low genetic variation, based on single copy gene analysis of the samples we collected within Kāneohe Bay are consistent with previous work. Concepcion and co-workers (*Concepcion, Baums & Toonen, 2014*) showed that the conserved nature of the local (O'ahu) population structure is likely a consequence of self-recruitment at the island scale. Low genetic variation may pose a barrier for rapid adaptive evolution in response

to climate change. However, the observed low levels of polymorphism could facilitate genome sequencing assemblies, mapping RNA-seq data, methylomics, proteomics, and microbiome analyses that enable studies of adaptive responses in local reefs that undergo different environmental stresses (*Guadayol et al., 2014*).

A more broadly sampled population study of *M. capitata* in Hawai'i has shown complex genetic histories of single coral colonies in the form of inter-species chimerism (*Work et al., 2011*). Our examination of the nature of host genetic variation, albeit based on a small number of colonies, provides evidence of historic inter-species chimerism only in the mitochondrial data. These remnants of past chimerism or hybridization may continue to contribute to the genetic toolkit of *M. capitata*. Here, the mtDNA is represented at low frequency (6% in DNA from a single egg sperm bundle) by the lineage of *M. flabellata*; a species known to form inter-species chimeras with *M. capitata* (*Work et al., 2011*). Because this variation was detectable within a single bundle from an individual polyp and across a colony, the mitochondrial complement likely contains significant variation; i.e., is heteroplasmic (*Rand, 2001*). The *M. flabellata* genome has apparently introgressed into the Kāneohe Bay *M. capitata* population although it appears that the nuclear genome regions we studied are free of *M. flabellata* DNA. Selection may remove foreign (nuclear) lineages from this *M. capitata* population (e.g., Montipora White Syndrome) although different mtDNA haplotypes, if differing only with respect to neutral DNA changes, have been retained in the heteroplasmic mitochondrial pool (*Aanen, Spelbrink & Beekman, 2014*; *Jacobsen et al., 2016*). It is also possible that different mtDNAs derived via introgression confer selective advantages (*Boratyński et al., 2011*) to coral physiology and can become fixed or maintained over time.

By taking a genome-wide approach, our findings provide unambiguous markers and genomic resources for further functional genomic studies of coral adaptation and acclimatization to ongoing environmental change. These results also lead to a more informed view of the role of coral genomics in coral holobiont adaptation in which the nuclear and mitochondrial compartments provide different perspectives.

## ACKNOWLEDGEMENTS

We thank the Rutgers School of Environmental and Biological Sciences for supporting the activities of the Genome Cooperative that generated much of the data used in this research.

### Funding

This work was made possible by grants from the National Science Foundation EF-1416785 awarded to P.G.F., T.M. and D.B, OCE-PRF 1323822 to H.M.P., and Hawaii EPSCoR EPS-0903833. Funding was also provided by the Paul G. Allen Family Foundation to R.D.G. and by the US-Israel Binational Science Foundation (BSF-2014035) to D.K.A. and T.M. The funders had no role in study design, data collection and analysis, decision to publish, or preparation of the manuscript.

## Grant Disclosures

The following grant information was disclosed by the authors:
National Science Foundation: EF-1416785.
US-Israel Binational Science Foundation: BSF-2014035.

## Competing Interests

The authors declare there are no competing interests.

## Author Contributions

- Hollie M. Putnam conceived and designed the experiments, performed the experiments, analyzed the data, contributed reagents/materials/analysis tools, wrote the paper, prepared figures and/or tables, reviewed drafts of the paper.
- Diane K. Adams conceived and designed the experiments, performed the experiments, contributed reagents/materials/analysis tools, reviewed drafts of the paper.
- Ehud Zelzion and Huan Qiu analyzed the data, contributed reagents/materials/analysis tools, wrote the paper, prepared figures and/or tables, reviewed drafts of the paper.
- Nicole E. Wagner performed the experiments, contributed reagents/materials/analysis tools.
- Tali Mass performed the experiments.
- Paul G. Falkowski reviewed drafts of the paper.
- Ruth D. Gates reviewed drafts of the paper, logistical support critical for making collections.
- Debashish Bhattacharya conceived and designed the experiments, analyzed the data, wrote the paper, prepared figures and/or tables, reviewed drafts of the paper.

## Field Study Permissions

The following information was supplied relating to field study approvals (i.e., approving body and any reference numbers):

M. capitata corals were collected on June 2015 (Special Activity Permit 2015-17) from Kāneohe Bay, O'ahu, Hawai'i.

## DNA Deposition

The following information was supplied regarding the deposition of DNA sequences:

The genomic data created for this project are available under the NCBI BioProject PRJNA339779.

## Data Availability

NCBI BioProject PRJNA339779.

## Supplemental Information

Supplemental information for this article can be found online at http://dx.doi.org/10.7717/peerj.3319#supplemental-information.

# PeerJ

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
