# Peer review of "Divergent evolutionary histories of DNA markers in a Hawaiian population of the coral Montipora capitata"

_PeerJ, doi:10.7717/peerj.3319_

## Round 0.1 · original submission · Major Revisions

I have now received the opinions of two reviewers. Both were generally positive about the submission, with one recommending acceptance as is, and the other minor revisions. However, having gone over the paper myself, I agree with reviewer 2 on many points, and feel that addressing the thorough and helpful concerns of the reviewer will require more work than a normal minor revision would entail; thus my decision is major revisions are needed.

In addition, I have two minor comments of my own from the paper on top of what the reviewers have already stated:

1. As already mentioned, please state your n=? more clearly in some sections; this is hard to discern in some sections. Additionally, as some sections include only very low numbers of specimens used, please include caveats in a bit more detail than given on line 339.

2. Line 342 - you state "the latter" but there is no "former" in the previous sentence.

I look forward to viewing your resubmission.

Reviewer 1 ·

Basic reporting

no comment

Experimental design

no comment

Validity of the findings

no comment

Additional comments

This manuscript by Putnam et al is a simple and straightforward work carried out to understand divergent evolutionary history of DNA markers in the coral Montipora. I think it will be an informative preliminary contribution to the future work related to studies of coral adaptation and acclimatisation using those marker

Reviewer 2 ·

Basic reporting

This study is deserve to be published in Peer J after revision.

Experimental design

I suggest the authors to consider and change experimental design, especially number and place of origin about samples which they use in this study, according to their study aims.

Validity of the findings

Finding is deserve to be published but after revision.

Additional comments

This study, entitled “Divergent evolutionary histories of DNA markers in a Hawaiian population of the coral Montipora capitata”, investigated intra and inter-colony sequence variation in a population of the dominant Hawaiian coral Montipora capitata by analyzing marker gene and genomic data. The manuscript is clearly written and unambiguous language. And the data from genetic analyses presented in this study are enough to search for the divergent evolutionary histories of several coral DNA markers, and I believe those data and methodologies are very informative.

However, discussion to support the study aims which authors set in advance is still not strong enough although you took enough data. And most of discussion seems to be ambiguous because discussion was not based on introduction and some results of the study. As results of them, I am confused what you aim in this study. What for did you investigate intra and inter-colony sequence variation? For development of informative methods for detailed analysis to distinguish inter-species chimerism of M. capaitata and M. flabellata leads to Montipora white syndrome (Work et al. 2011)? Because dispersive descriptions about M. flabellata in relation to MWS occurred in the MS. Or, to elucidate the difference of frequencies of chimera and/or mosaicism in detail of M.capitata as to colony and/or reefs? So I strongly suggest you to define your study aims, and after that, change both introduction and discussion for coincidence between them.

And numbers of colonies and bundles you used in this study are not enough. Especially, it seems that Mcap4 and Mcap5 were not used in any analyses. The sentence, “Analysis of Pax-C intron PCR products from pooled bundles from all five colonies (i.e., including Mcap4 and Mcap5) also showed a single SNP in this region.” was shown in line 248-249, however, I did not find results including Mcap4 and/or Mcap5 in any figures and tables, even in Table S1 which showed naming scheme used for the Montipora capitata samples used in this study. These mean that numbers of colonies you used are three, are these right? Three colonies just may be enough for any phylogenetic analyses, because we can use other sequences of target species from any database. Whereas if you hope to elucidate the intra colony genetic structure in M. capitata, you need to perform single-copy gene analysis to confirm that the genomic-based SNP data from one bundle match exactly the multiple bundle sperm PCR output from same colony, not only in Mcap2 but also Mcap1 and Mcap3. I apologize above claims if authors already had performed them using analysis shown in Line 320-323.


Minor points

Line 94-95: with a natural photoperiod and ambient seawater conditions.

How condition is “natural” and “ambient”? Show records of any environmental parameters in outdoor tanks, if you recorded them.


Line 323: This result suggests that Mcap1-3 from Reefs 19 and 51 are closely related.

I do not know whether these reefs are close or not because scale bar was not shown in Figure 1. Show the bar in this figure. And you need to show some reason why you did not find no evidence of novel SNPs in genes from Mcap1 and Mcap3 in Discussion. You may refer to the proceeding study about population genetics and/or connectivity of Montipora in this area, if you have data by yourself and/or some references.


Line 339-342: These results, albeit based on a small number of colonies, provide a counterpoint to more broadly sampled population studies that show complex genetic histories of single coral colonies, including inter-species chimerism of M. capitata and M. flabellata (Work et al., 2011).

I think this sentence is out of line. As mentioned above, number of colonies which you used in this study were only three (or five?) and only two reefs. Clarify this sentence.


Line 351-354: Selection may remove foreign (nuclear) lineages from this M. capitata population (as for MWS) although different mtDNA haplotypes, if differing only with respect to neutral DNA changes, have been retained in the heteroplasmic mitochondrial pool.

I do not understand what this sentence means, and this may serve your interest. Show some references (not apply only to corals) about the relationship between removal of nuclear lineages and retaining of mtDNA haplotypes, if you have.


Figure 5: Is Figure 5A) described in text? Remove them if this explanation are not needed to the MS.

---

## Round 0.2 · accepted · Accept

The revision is well done - I have made only a few small corrections on the attached PDF; please see these are done at the proof stage or earlier if possible.

I look forward to seeing the published version of your work.

Reviewer 2 ·

Basic reporting

This study is deserve to be published in Peer J.

Experimental design

Experimental design is deserve to be published by revision.

Validity of the findings

Finding is deserve to be published.

Additional comments

Overall, this manuscript was adequately revised according to comments by reviewer and editor, and I can recommend it for publication in PeerJ.
Check again spell errors and missing of any references before publication.